# Underrated Innovativeness of Micro-Enterprises Compared to Small to Medium Enterprises in the Slovenian Forest-Wood Sector

Ana Slavec 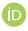

InnoRenew CoE, Livade 6, 6310 Izola, Slovenia; ana.slavec@innorenew.eu

**Abstract:** Although micro-enterprises represent most of the enterprises across different sectors, they are excluded from official statistics on innovation activities. What we know about micro-enterprises is based on smaller quantitative and qualitative studies that are country- and sector-specific. To understand the innovation activities of Slovenian enterprises in the forest-wood sector, we conducted our own quantitative study in 2019 based on the Eurostat's Community Innovation Survey (CIS) questionnaire. Based on responses from 294 enterprises, we compare how micro-enterprises and small to medium enterprises (SMEs) differ in innovation strategies, product, and process innovations, co-operation with other organisations, innovation activities, and innovations with environmental benefits. The results indicate that, in some respects, enterprises with two to nine employees are at least as innovative as small to medium enterprises, or even more so. We argue that innovation surveys should lower the employee count threshold to attain better representative insight into the innovation landscape.

**Keywords:** micro-enterprises; innovation activities; forest-wood sector; innovation surveys

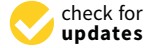



## 1. Introduction

Slovenia is a country that lags in innovation, which is particularly true for its furniture industry and other industries that are part of the forest-wood value chain. A steep decline in revenues after the 2008 financial crisis led to the closure of many large companies in this sector that has traditionally been very important for this small country with a forest coverage of almost 60%, which makes it the third most forested country in Europe [1].

The national Smart Specialisation Strategy [2] identified this sector as having a strong potential for growth, and it is believed that leveraging innovation is of key importance in supporting its recovery process. Moreover, innovations in the forest sector can potentially have an environmental aspect [3].

To develop a framework to support innovation, a deeper understanding of existing innovation activities, and the reasons behind the lack of them, is needed. However, innovation research linked to the forest sector usually lacks more advanced experimental designs and quantitative methods [4]. Another limitation is that more than 90% of this sector is comprised of enterprises with less than ten employees, i.e., micro-enterprises, which are usually excluded from official business surveys, such as the Community Innovation Survey (CIS).

That is not an issue only when studying the forest-wood sector but also for studying innovation in general. According to data from the OECD [5], micro-enterprises represent between 70% and 95% of all enterprises in all countries, but it can be even higher for some sectors. For instance, in the Spanish tourism sector, 96% of enterprises have less than ten employees [6].

Since micro-enterprises are often under-surveyed or completely excluded in most surveys on the topic of innovation, much remains unknown about their innovation activities [7] and how organisational capabilities can be developed in enterprises of this size [8]. Small

and micro-enterprises are also under-represented in research concerning environmental performance and sustainability innovation [9]. The purpose of this paper is to study innovation in micro and small to medium enterprises (SMEs), focusing on how enterprise size impacts on business strategy, innovation adoption, innovation activity, co-operation with other enterprises or organisations, and innovations with environmental benefits, especially in the forest-wood sector. While most innovation research focuses on larger enterprises, this paper fills the gap regarding innovation features of smaller enterprises, particularly micro-enterprises.

The research question this study addresses is how micro-enterprises compare to SMEs in Slovenia's forest-wood sector in terms of innovativeness. After reviewing the existing literature on micro-enterprises and SMEs in different sectors and formulating a hypothesis about the characteristics of their innovations and innovation activities, the results of a cross-sectional survey about innovation in the Slovenian forest-wood sector that used selected questions from the CIS are presented and discussed.

## 2. Literature Review

CIS is a biannual survey that provides statistics about different innovations and innovation activities for enterprises in most countries that are part of the European statistical system. Although the CIS sample only includes enterprises with ten or more employees, in practice, it can happen that they include enterprises with a lower number of employees than the actual information in the registry. Thus, there are a few studies of micro-enterprises based on secondary analysis of the CIS data. For instance, in Croatia, it was found that micro-enterprises in the CIS are less likely to innovate than small to medium enterprises but are more likely to innovate than large enterprises [10]. A similar analysis based on CIS data was done for the Czech Republic's manufacturing industry, and the results show that larger enterprises have more process innovations and fewer research and development expenses per employee [11]. In contrast, a study of Greek enterprises in the CIS found a weak negative correlation between size and innovation performance [12]. However, a UK study found no significant association between enterprise size and innovative sales after multiplying the share of innovative sales with turnover and dividing it by the number of employees to account for extreme variables in single-product enterprises [13].

Nevertheless, the number of micro-enterprises for which data is available through the CIS is limited and is not representative of the category. In the following subsections, the study's theoretical framework is presented, structured according to selected question topics in the CIS. Moreover, hypotheses on different innovation aspects of micro enterprises and SMEs are developed based on the literature, including findings of several previous studies on the topic that included primary data collection, both qualitative and quantitative.

### 2.1. Business Strategy

According to Porter [14] there are three generic strategies for how an enterprise pursues a competitive advantage in a market, either by low cost, differentiation relative to its rivals (higher quality) or focusing on one or few segments instead of the whole industry. Presuming that large enterprises have more efficient facilities, distribution systems, service organisations and other functional units for their size, they have a cost advantage over smaller enterprises [14]. Thus, it is easier for an enterprise with more employees to pursue differentiation and overall cost leadership, while we can assume that those with fewer employees focus only on a particular segment.

This is in line with findings from a survey of suppliers to the UK public sector that found that smaller enterprises have more difficulties with procurement processes that are an important driver of innovation [15]. Moreover, a qualitative study of micro business owners at an Indonesian university that focused on barriers to innovation found that they often lack the necessary human resources and capital [16]. Next, a Slovakian survey highlighted the lack of marketing departments among micro-enterprises [17] and a survey of small and micro-enterprises in Austria and Germany found them to be disadvantaged

compared to large enterprises due to their limited financial and other resources and lack of knowledge; however, it emphasized their potential, because of strong customer-orientation and openness to new ideas [18]. Similarly, a Polish survey found that smaller enterprises pay more attention to customer needs [19] and another UK study found that, in remote rural areas, more micro-enterprises and SMEs use innovation based on understanding the needs of a customer base as a strategy [20].

Due to the lack of resources and orientation to customer needs, H1 is suggested: compared to SMEs, micro-enterprises place less importance on strategies, such as improving existing products (H1a), introducing entirely new products (H1b), reaching new customer groups (H1c) and low-price (H1e), while they put more importance on customer-specific solutions (H1d).

### 2.2. Adoption of Different Types of Innovation

According to the basic definitions in the Oslo Manual [21], which presents guidelines for collecting and interpreting innovation data, innovation is "the implementation of a new or significantly improved product (good or service), or process, a new marketing method, or a new organisational method in business practices, workplace organisation or external relations". Innovations can be developed by the enterprise itself or they can be developed in co-operation with other enterprises or institutions or mainly by other enterprises or institutions.

Enterprise size is often mentioned as a positive predictor in research on innovation adoption but is often without a consistent definition of the construct [22,23]. In most studies included in this review it is measured as the number of employees, which is in line with the Oslo Manual [21]. For instance, one of the earliest studies that showed a linear positive relationship between firm size and the number of patented inventions was on larger firms in Sweden; however, it did not include smaller enterprises [24]. On the other hand, a study on the Spanish manufacturing sector found that the relationship is not necessarily linear [25] and some of the secondary analyses of CIS data have found a negative association [12] or no association at all [13]. There are also two studies of innovation activities in the forest sector, one Central European [26] and one North American [27], that have collected data on enterprise size but did not use these data in the analysis and so, unfortunately, no conclusions could be drawn from these data.

Although the evidence on the association between innovation is somewhat inconclusive, most studies indicate a positive association between enterprise size and innovation adoption [22,24]. Thus, H2 is proposed: among micro-enterprises there are fewer innovations in goods (H2a), services (H2b), production processes (H2c), distribution methods (H2d) and supporting activities for processes (H2e) than among SMEs.

As we will see in Section 2.4, micro-enterprises depend on co-operation with other enterprises and organisations [28–30], and thus H3 is proposed: compared with SMEs, micro-enterprises introduced fewer innovations done by themselves (H3a) and more together with other enterprises or organisations (H3b) and by adapting or modifying processes originally developed by other enterprises or organisations (H3c) and completely by other enterprises or organisations (H3d).

On the one hand, a survey focused on the forest-based bioeconomy in the EU found that due to the centralisation of decision-making, smaller companies tend to be more flexible, which makes them better able to develop newer and more radical innovations [31]. On the other hand, a study on innovation in forestry in Central Europe found only innovation in the form of introducing those innovations originally developed by others and none that would be completely new to the market [26]. A Slovakian survey that focused on eco-innovations among micro and small enterprises found that innovations are usually only new to the enterprises and not new to the sector [3]. This is more in line with the previous hypotheses about who developed the innovations that were introduced. Consequently, we propose H4: product innovators among micro-enterprises are less likely than innovators

among SMEs to introduce goods or services that are new to their market (H4a) and more likely to introduce those that are new to their enterprise (H4b).

### 2.3. Innovation Activities

The Oslo Manual guidelines [21] define innovation activities as "all scientific, technological, organisational, financial and commercial steps which actually lead, or are intended to lead, to the implementation of innovation". Moreover, during a given period, innovation activities might be successful (i.e., resulting in the implementation of a new innovation), ongoing (work in progress) or abandoned before the implementation of an innovation.

Because of their limited resources [16,18], we propose H5: micro-enterprises are more likely than SMEs to have had innovation activities that did not result in product or process innovations because the activities were abandoned or suspended before competition (H5a) or were still ongoing (H5b).

Similarly, because they usually do not have their own research and development and marketing departments [11,17] and have less resources [16,18], H6 is proposed: compared to SMEs, micro-enterprises have fewer in-house research and development innovation activities (H6a), more external research and development (H6b), less acquisition of machinery, equipment, software and buildings (H6c), more acquisition of existing knowledge from other enterprises or organisations (H6d), more training for innovative activities (H6e), less market introduction of innovations (H6f) and fewer innovation activities in design (H6g).

### 2.4. Co-Operation with Other Enterprises or Organisations

A quantitative survey studied enterprises in the agri-food sector in the Campania region of Italy and found that public funding, particularly in the form of innovation networks and collaborations with universities and research institutes, plays an important role in advancing the innovation capacity of medium, small, and micro-enterprises [29]. Another Italian survey focused on the manufacturing sector in the Piedmont region and confirmed that openness to collaboration leads to better opportunities [32]. Similarly, a study on small food manufacturers in six European regions found that their workforce usually lacks internal expertise, and their innovation activities depend on collaboration with research institutes [28]. The importance of having a collaborative innovation strategy was also stressed in a qualitative study of student micro owners at an Indonesian university [16]. Enterprises that do not have the capacity to innovate by themselves can benefit from using living labs, as indicated from the results of a Swedish survey [33]. In addition, a Spanish survey study found that collaboration is particularly important in knowledge-intensive industries, such as biotechnology [30].

However, due to lower research and development capital, smaller firms have a lower ability to absorb knowledge from science-oriented sources but are more successful in using generally accessible knowledge from customers, suppliers, trade journals and conferences [34]. Correspondingly, a Slovenian survey on open innovation found that micro-enterprises collaborate with knowledge institutions (i.e., universities and research institutes) and consultancy companies less often than SMEs, while there are no statistically significant differences in collaboration with customers, suppliers, competitors, and other companies [35]. Another Slovenian study on open-innovation found that companies that are not engaging in this kind of activity are smaller on average [36].

As argued above, innovation activities in micro-enterprises depend more on co-operation with others. However, they are less likely to be a part of an enterprise group and to have science-oriented co-operations. Thus, H7 can be assumed: compared with SMEs, micro-enterprises have less co-operation with other enterprises in their enterprise group (H7a), more co-operation with suppliers of equipment, materials, components, or software H7b), clients and customers in the private (H7c) and public sector (H7d), competitors and other enterprises in their sector (H7e), less co-operation with consultants or commercial labs (H7f), universities or other higher education institutes (H7g) and government, public, or private researcher institutes (H6h).

*2.5. Innovations with Environmental Benefits*

For SMEs and micro-enterprises, declaring socio-environmental footprint sustainability represents a higher cost than for larger enterprises [37]. A survey of micro-enterprises in the Italian craft beer industry indicates a strong influence of environmental awareness, a weak influence of external pressures on proactive environmental strategies and no effect for internal drivers [9]. Furthermore, a survey on eco-innovations in the Slovak forestry service found no strategic eco-innovators, only eco adopters, and a push from the outside environment is needed to make them more active [3].

Lastly, based on their lack of resources, we propose H8: innovators among micro-enterprises have introduced fewer innovations with environmental benefits than SMEs. Specifically, less have reduced material and water use (H8a), reduced energy use or $CO_2$ footprints (H8b), reduced air, water, noise, or soil pollution (H8c), replaced a share of materials with less polluting or hazardous substitutes (H8d), replaced a share of fossil energy with renewable energy sources (H8e), recycled waste, water, or materials for their own use (H8f), facilitated recycling of product after use (H8g) and extended product life through longer-lasting, more durable products (H8h).

**3. Methods**

In 2019, we conducted a self-administered cross-sectional survey among Slovenian enterprises in the forest-wood sector. The questionnaire was composed of selected questions from CIS, but unlike the CIS, it included enterprises of all sizes, including micro-enterprises. The survey design followed guidelines for collecting and interpreting innovation data outlined in the Oslo Manual [21] and used a mixed-mode approach that combined a paper-and-pencil postal survey with online data collection.

Since the CIS 2018 questionnaire [38] was not available at the beginning of 2019 when our survey started, we used a selection of questions from the CIS 2016 questionnaire [39] and one question from the CIS 2014 questionnaire [40], using 2016 to 2018 as a reference period instead of 2014 to 2016 (CIS 2016) or 2012 to 2014 (CIS 2014). The CIS 2016 questions that we included are 1.4 (strategies), 2.1 to 2.4 (product innovation), 3.1 to 3.2 (process innovation), 4.1 (ongoing or abandoned innovation activities), 5.1 (innovation activities), 7.3 (co-operation for product and process innovations), 15.3 (average number of employees) and a few other questions that are not included in this analysis. The CIS2014 question we used is 13.1 (innovations with environmental benefits). We slightly adapted the order of questions and added some extra questions that we developed on our own, but we do not include them in this analysis (see Supplement S1). Before being fielded, the paper questionnaire was pre-tested on five enterprises to assess its clarity and improve the wording of questions and instructions.

The population were active enterprises in selected categories according to the Statistical Classification of Economic Activities in the EU (NACE) [41] that were created before January 2016. The sample was prepared based on the bizi.si registry of Slovenian businesses from which we retrieved the list of enterprises whose main activity was in one of the following seven categories according to the NACE standard: forestry and logging (A2); manufacture of wood and of products of wood and cork, except furniture; manufacture of articles of straw and plaiting materials (C16); manufacture of paper and paper products (C17); manufacture of furniture (C31), except for manufacture of mattresses (C31.3); other manufacturing (C32), except for manufacture of medical and dental instruments and supplies (C43.5); construction of buildings (F41); and wholesale of wood, construction materials and sanitary equipment (G46.3). After excluding non-active enterprises, those that are bankrupt or in the liquidation process, agrarian communities, associations, and interest groups, the sample frame included 7123 enterprises.

Expecting a low response rate, we decided to send the invitation to the full population. Data collection started on 15 January 2019, when the first mailing was sent, including the cover letter (Supplement S2), the printed questionnaire, an information sheet with frequently asked questions (Supplement S3) and a postage-paid return envelope. The cover letter included a link to an online questionnaire administered through the LimeSurvey platform. Enterprises could decide between responding on paper and sending it with the enclosed envelope or responding online by typing the link. After one month we sent an additional postal reminder to enterprises with an e-mail in the registry (32% of all enterprises on the list), followed by additional reminders if necessary (up to four e-mail reminders).

Data collected by paper surveys were entered in the online questionnaire and downloaded in CSV format. After removing incomplete responses and running consistency checks the data were imported to the SPSS statistical programme (Version 28) used to store and analyse these data, except for charts drawn using Microsoft Excel. The association between selected variables and enterprise size (grouped in three categories) was examined by crosstabulations and computing the Chi-squared test with a significance threshold of 0.05. The data and code can be accessed via the Slovenian Social Science Data Archive [42].

## 4. Results

In the nine months that the survey was active, we collected 294 completed responses; 336 units were found ineligible (not in business, bankrupt, liquidated, changed activity, etc.), 262 explicitly refused participation and 6233 did not answer. Almost three in four (74%) questionnaires were completed on paper, while the remaining quarter (26%) responded online. Based on recommendations from the AAPOR [43], the response rate was calculated as a ratio of completed questionnaires with the total sample without units that are not eligible and ranged from 2.4% for the construction sector to 6.8% for the wood manufacturing sector.

Among the respondents, 110 (39%) were micro-enterprises with 0–1 employee, 106 (37%) with 2–9 employees, 43 (15%) had ten to less than 50 employees (small), 20 (7%) had 50 to less than 250 employees and only three (1%) had more than 250 employees (large). The latter were excluded from the analysis by enterprise size, while small and medium enterprises were merged into one category (SME). The following figures compare selected CIS indicators between the two micro-enterprise categories and SMEs (for detailed numbers, see Table S1 in Supplement S4).

### 4.1. Business Strategy

The degree of importance of different strategies for enterprises was measured on a 4-point scale (high, medium, low, not important) that we recoded into a dummy variable (1—high, 0—other). The most important strategies are improving existing products (44.6% high) and customer-specific solutions (39.1% high), but there are differences according to enterprise size.

As indicated by Figure 1, fewer respondents give high importance to the strategy of improving existing products among micro-enterprises with 0–1 employee (31.1%) than both among micro-enterprises with 2–9 employees (57.0%) and SMEs (54.7%; $\chi^2$ = 16.5, $p < 0.01$). A significant difference ($\chi^2$ = 7.7, $p = 0.02$) was also found for reaching new customer groups; both micro-enterprises with less than two (19.0%) and more employees (26.0%) have a lower share of high importance than SMEs (38.5%). A similar difference between different enterprise sizes can be observed for customer-specific solutions, but it is not statistically significant and there is an even smaller difference for introducing entirely new products and low-price strategies.

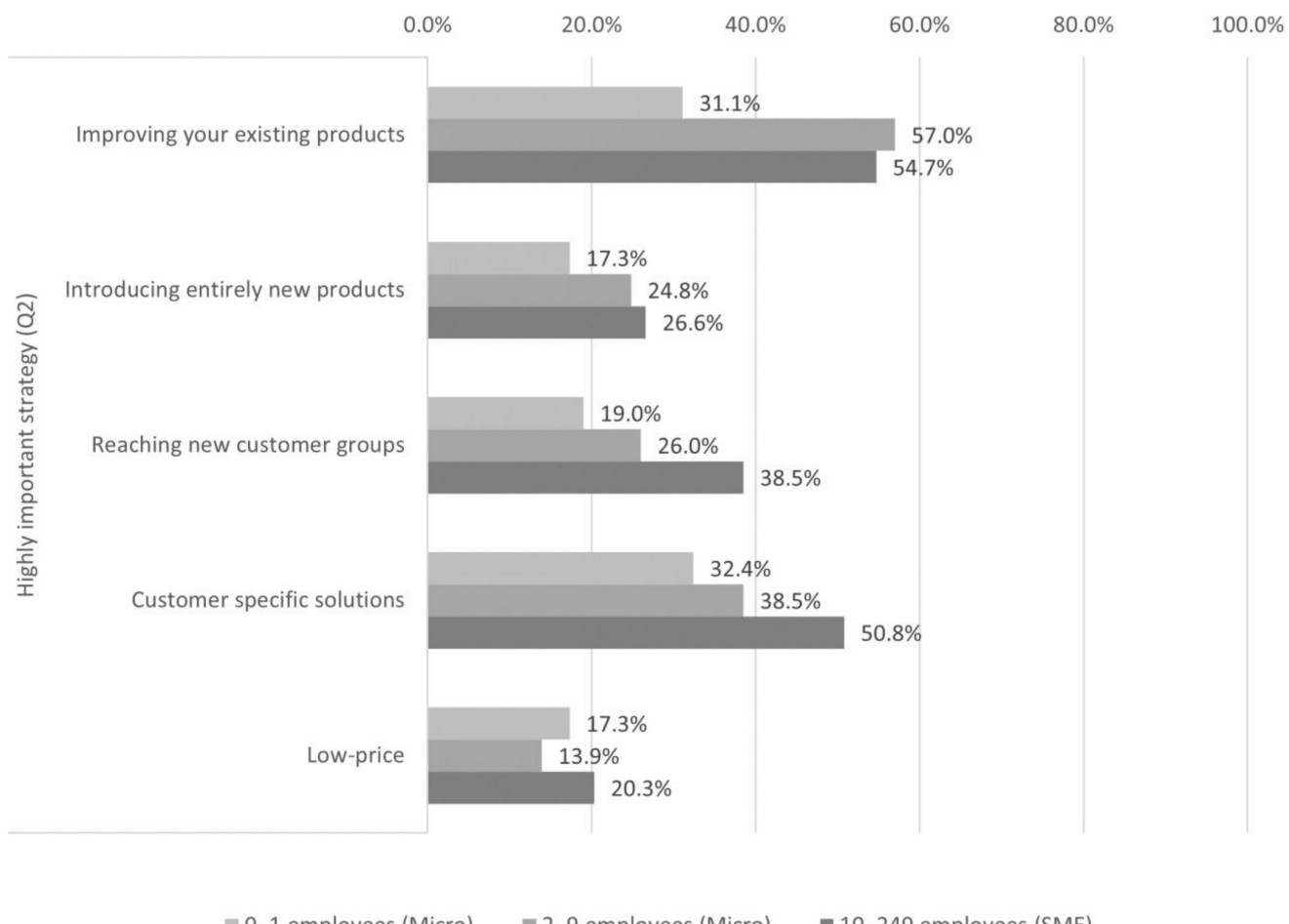

**Figure 1.** Responses to the question: During the three years 2016 to 2018, how important were each of the following strategies to your enterprise?

### 4.2. Adoption of Different Types of Innovation

In total, 40.4% of enterprises have introduced new or significantly improved goods, 35.1% services, 38.7% production processes, 22.3% distribution methods and 27.1% supporting activities for processes. Except for production process innovations, statistically significant differences have been found in all other product and process innovations (Figure 2). More enterprises have produced new or significantly improved goods among SMEs (44.3%) and even more among micro-enterprises with 2–9 employees (47.7%) than micro-enterprises with 0–1 employee (31.2%; $\chi^2 = 6.6$, $p = 0.04$). Similarly, innovations in distribution methods are more frequent among micro-enterprises with 2–9 employees (30.2%) than among both SMEs (21.7%) and micro-enterprises with 0–1 employee (15.0%; $\chi^2 = 6.6$, $p = 0.04$). For service innovations, the share is lower among SMEs (24.6%) compared to both micro-enterprises with 0–1 (29.9%) and 2–9 (46.3%) employees ($\chi^2 = 10.2$, $p < 0.01$). In contrast, new or significantly improved supporting activities for processes were significantly more frequent among SMEs (45.9%) than among micro-enterprises with 2–9 (29.9%) and especially 0–1 (13.0%) employee ($\chi^2 = 21.4$, $p < 0.01$).

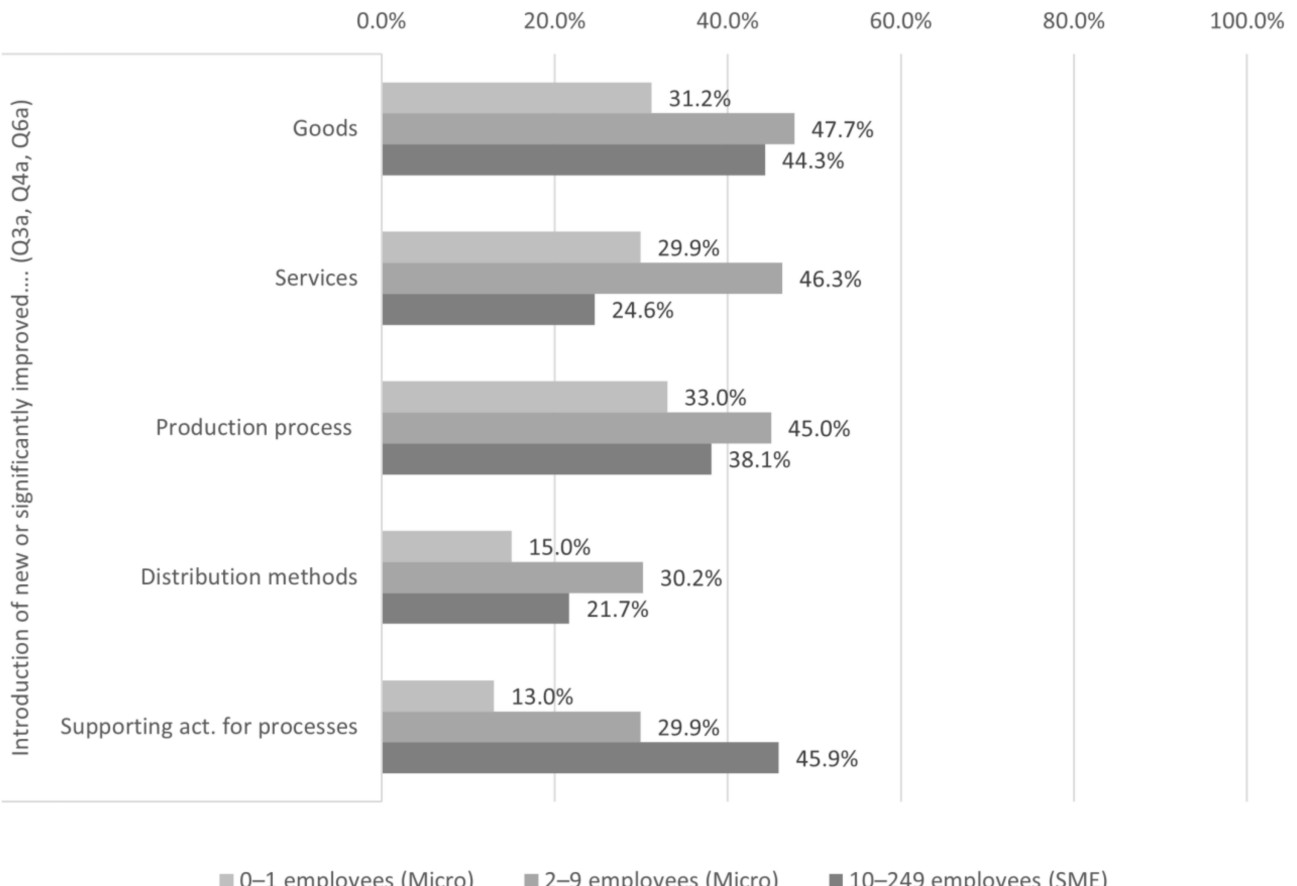

**Figure 2.** Responses to the question: During the three years 2016 to 2018, did your enterprise introduce new or significantly improved goods/services/ . . . ? (Tick all that apply).

Enterprises that have introduced innovated goods, services, or processes were asked who developed them and multiple answers were possible. For all types of innovations, the most common response was that they were introduced by the enterprise itself: 81.6% for goods, 74.4% for services and 71.5% for processes, followed by "your enterprise together with other enterprises or organisations" (52.5% for goods, 57.4% for services and 52.4% for processes). The third most popular response for service and process innovation was "other enterprises or organisation" (30.6% for services and 36.6% for processes), while for goods innovations, it was "your organisation by adapting or modifying processes originally developed by other enterprises or organisations" (30.8%).

The only significant difference among differently sized enterprises has been found for the share of service innovations that were done by adapting or modifying processes originally developed by other enterprises or organisations ($\chi^2 = 6.1$, $p = 0.05$). Specifically, more service innovations were implemented this way among micro-enterprises with 2–9 employees (42.1%) than SMEs (15.4%) and micro-enterprises with 0–1 employee (7.1%) (Figure 3). For innovations done by the enterprises themselves or together with other enterprises or organisations or totally by other enterprises or organisations, there are no significant differences by number of employees.

Product innovators were asked if any goods or services innovations were new to their market (they may have already been available to other markets) and if any were only new to their enterprise (available from their competitors on their market). In total, 53.4% of product innovators have introduced products that were new to their market and 66.7% introduced products that were only new to their enterprise. There are no significant differences among micro-enterprises and SMEs for the two indicators (Figure 4).

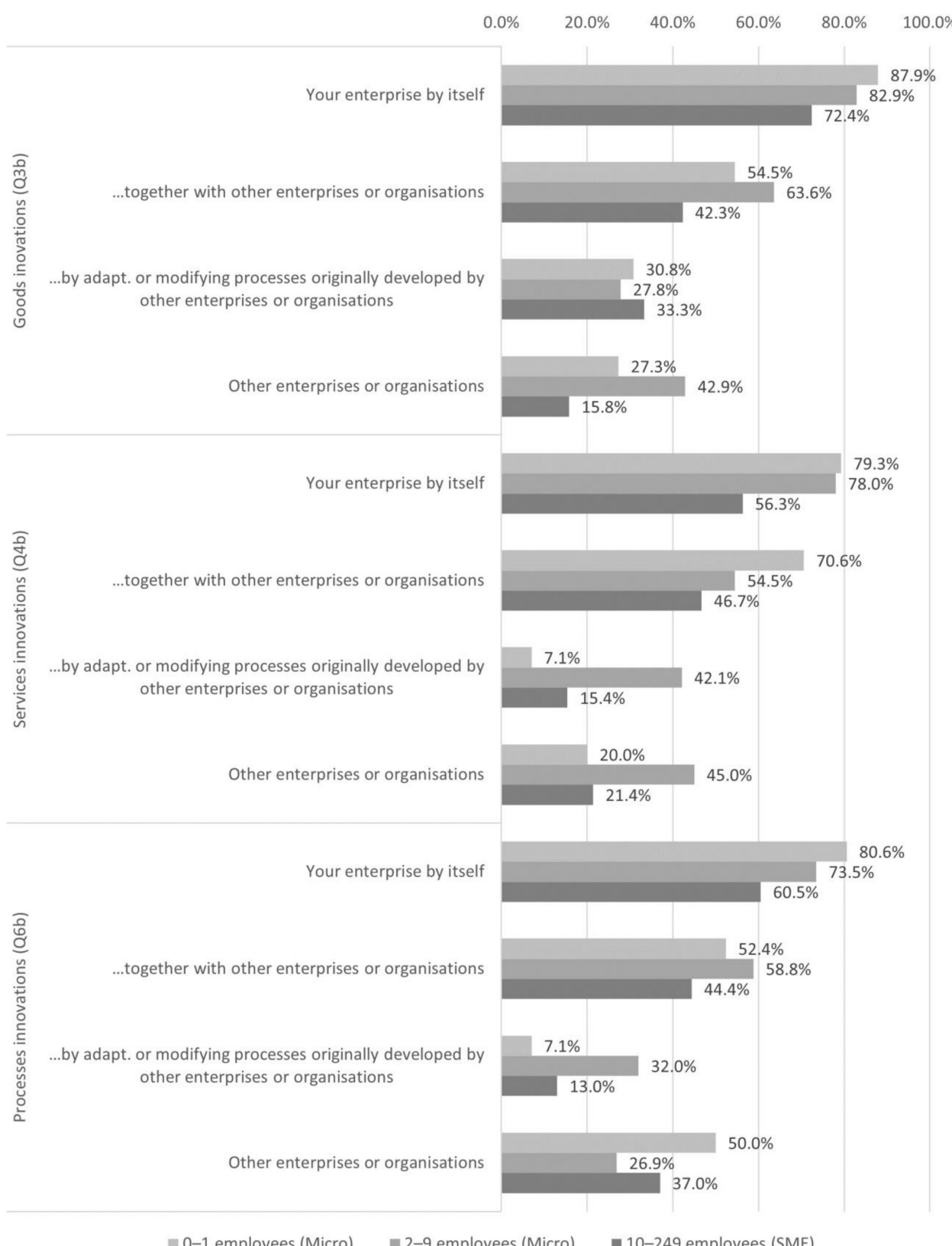

**Figure 3.** Responses to the question: Who developed goods/services/processes that the enterprise introduced during the three years 2016 to 2018?

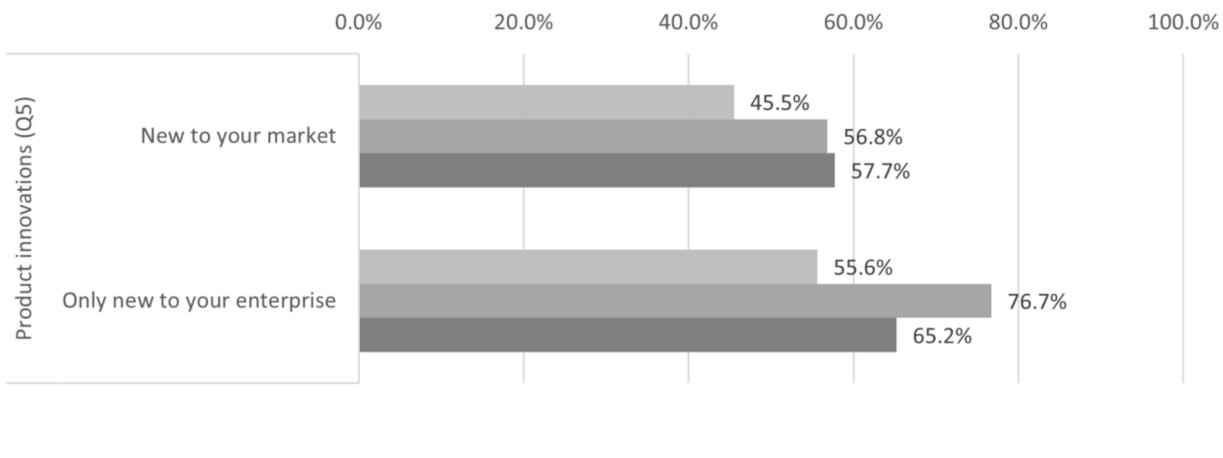

**Figure 4.** Responses to the question: Were any of your product innovations (goods or services) during the three years 2016 to 2018 new to your market or only new to your enterprise?

*4.3. Innovation Activities*

All enterprises were asked if they had any innovation activities that did not result in product or process innovations because the activities were abandoned or suspended before competition or were still ongoing at the end of 2018. In total, 11.5% of enterprises had abandoned or suspended innovation activities, and 16.8% still had ongoing innovation activities. No significant differences between micro-enterprises and SMEs were found for either indicator (Figure 5).

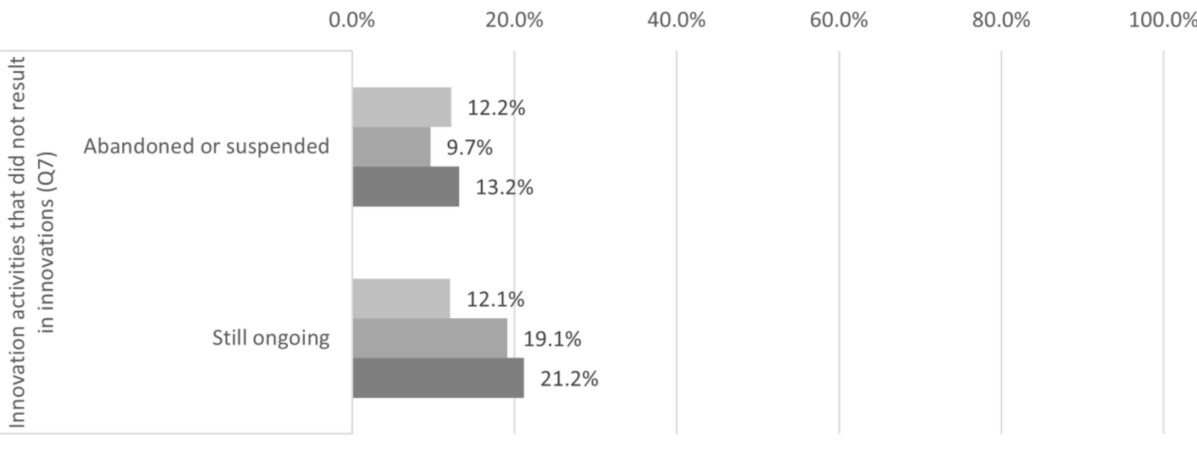

**Figure 5.** Responses to the question: During the three years 2016 to 2018, did your enterprise have any innovation activities that did not result in a product or process innovation because the activities were abandoned or suspended/still ongoing?

Enterprises that responded positively to at least once for either one of the questions about innovations (Q3, Q4, Q6) or the question about innovation activities (Q7) were asked about the types of innovation activities their enterprise engaged in and multiple responses were possible. The most popular response was acquisition of machinery, equipment, software, and buildings (86.5%), followed by in-house research and development (69.5%),

design (48.4%), acquisition of existing knowledge from other enterprises or organisations (47.1%), external research and development (42.0%), training for innovative activities (34.9%) and market introduction of innovations (25.0%). As indicated in Figure 6, there were no significant differences between micro-enterprises and SMEs for any of the listed innovation activities.

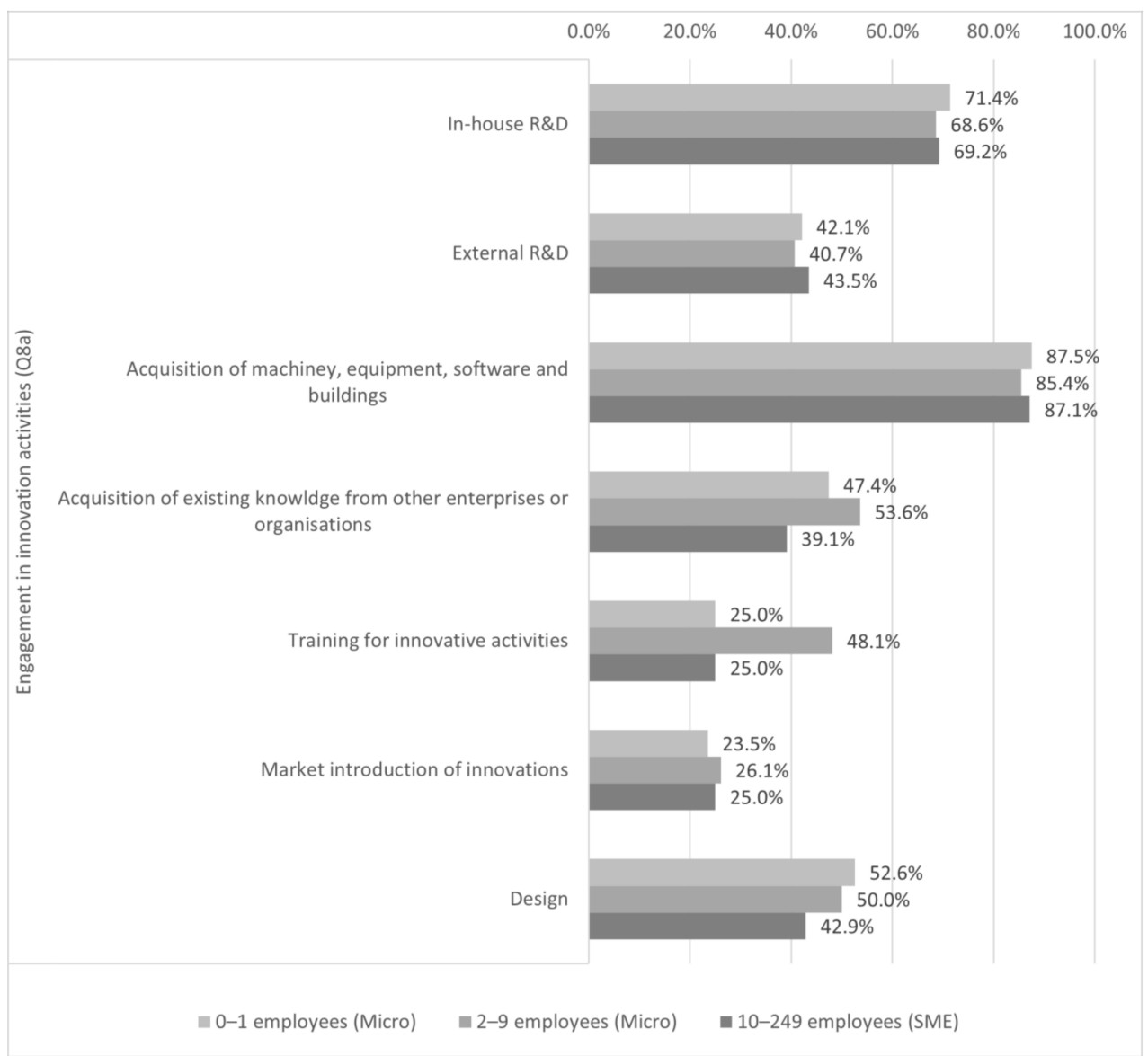

**Figure 6.** Responses to the question: During the three years 2016 to 2018, did your enterprise engage in the following innovation activities? (Tick all that apply).

### 4.4. Co-Operation with Other Enterprises or Organisations

The same enterprises have been asked with what kind of other enterprises and organisations did they co-operate in their innovation activities. The question was asked both for Slovenia and internationally, but in the results, we merged the responses. Most enterprises co-operated with suppliers of equipment, material, components, or software (82.1%), while only 53.8% co-operated with clients or customers from the private sector, 44.0% with other enterprises within their enterprise group, 25.4% with competitors and other enterprises in their sector, 20.7% with clients or customers from the public sector, 16.7% with consultants or commercial labs, 15.0% with universities or other higher education institutes and 12.5% with government, public, or private researcher institutes. The only significant difference

between enterprises of different sizes (Figure 7) was found for suppliers of equipment, materials, components, or software ($\chi^2 = 8.5$, $p = 0.01$). There are more SMEs (91.2%) with these types of co-operators than micro-enterprises with 2–9 (86.8%) and 0–1 (66.7%) employees.

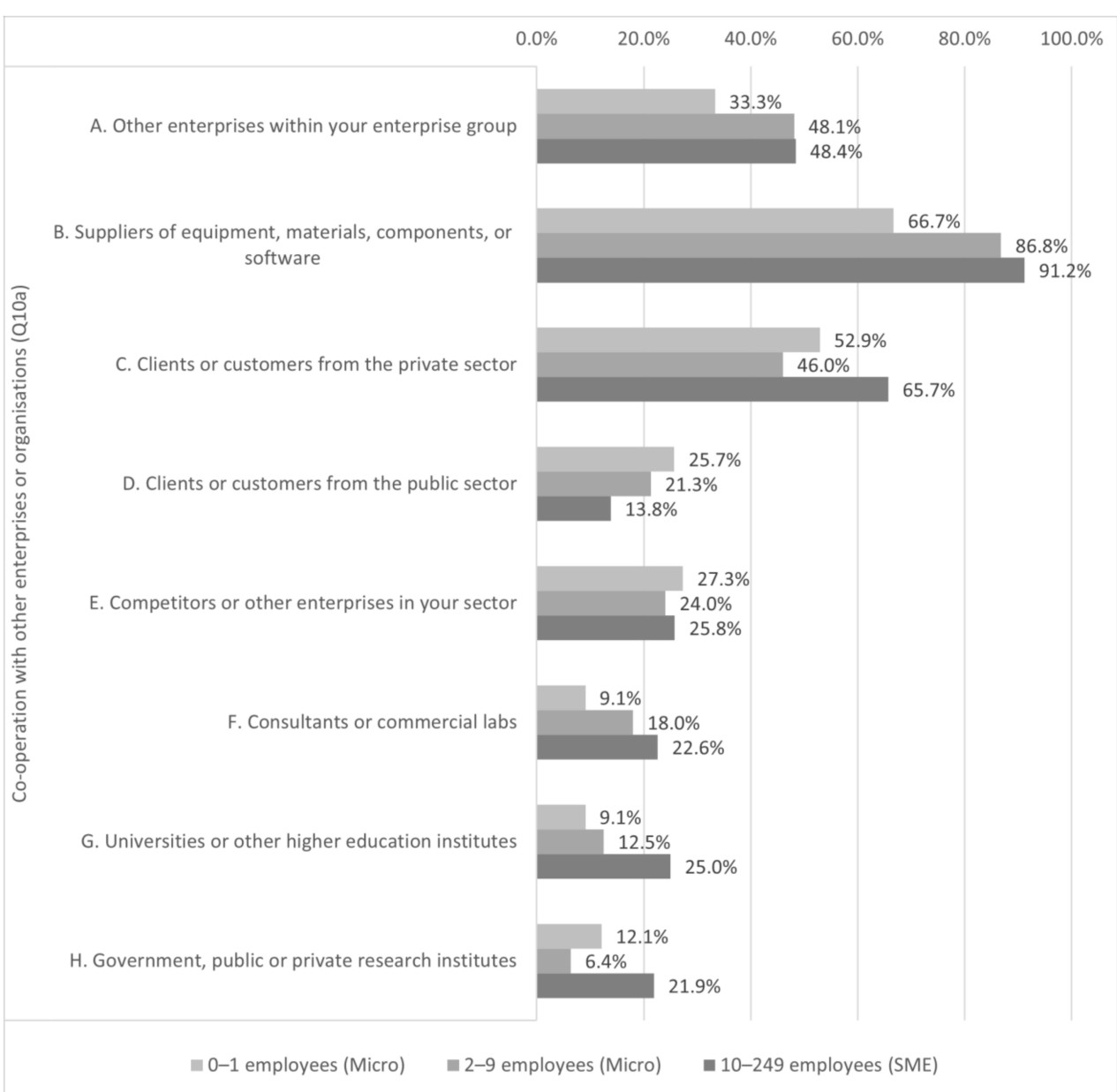

**Figure 7.** Responses to the question: During the three years 2016 to 2018, did your enterprise co-operate on any of your innovation activities with other enterprises or organisations?

*4.5. Innovations with Environmental Benefits*

Enterprises that introduced any type of product (Q3, Q4), process (Q6) or organisational or marketing innovations (Q11) were asked if any of these had any environmental benefits. Almost half of the responding enterprises reduced air, water, noise, or soil pollution (47.9%) and reduced energy use or $CO_2$ footprints (47.9%), while only 38.1% extended product life through longer-lasting, more durable products, 36.4% recycled waste, water, or materials for own use or sale, 32.8% replaced a share of materials with less polluting or hazardous substitutes, 31.9% reduced material and water use per unit of output, 31.0% facilitated recycling of product after use, and 20.3% replaced a share of fossil energy with renewable energy sources.

A significant difference between micro-enterprises and SMEs was found for two types of environmental benefits (Figure 8). While two-thirds (66.7%) of SMEs have reduced energy use or CO2 footprints, fewer enterprises with 2–9 (45.1%) and 0–1 (28.6%) employees have introduced innovations with this kind of environmental benefit ($\chi^2$ = 10.0, $p < 0.01$). In contrast, more than half of enterprises with 2–9 (52.0%) employees have introduced innovations that benefited in recycled waste, water, or materials for their own use or sale, compared to about a third (34.4%) of SMEs and about one-sixth of micro-enterprises with 0–1 employee (16.7%; $\chi^2$ = 11.4, $p < 0.01$).

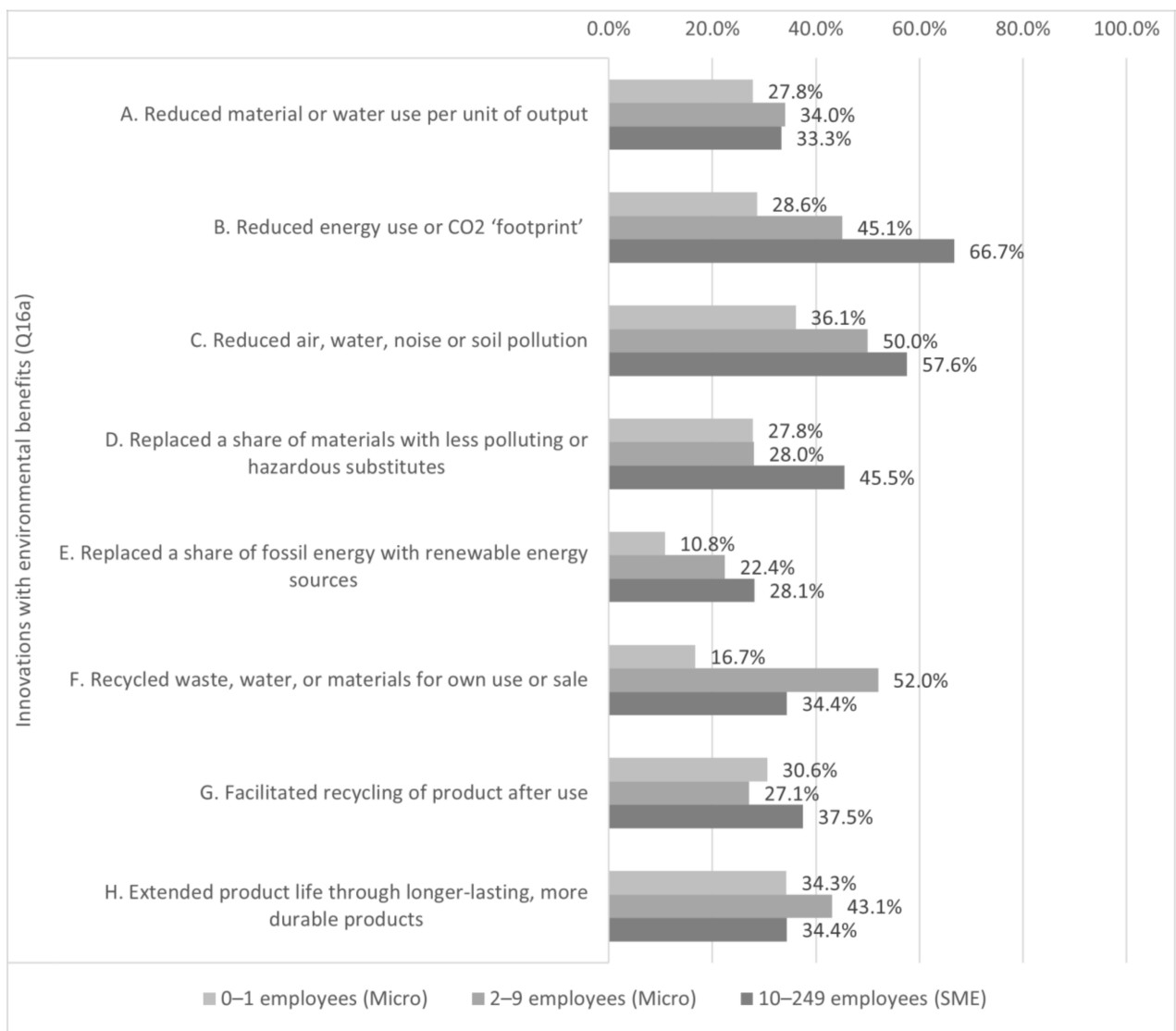

**Figure 8.** Responses to the question: During the three years 2016 to 2018, did your enterprise introduce product, process, organisational or marketing innovations with any of the following environmental benefits?

While the results do not confirm differences between micro and small to medium enterprises in the engagement in research and development and other innovation activities, there are statistically significant differences in the importance of four out of the five strategies, four out of five types of innovation adoption, one out of the eight types of co-operation partner and two out of the eight types of innovations with environmental benefits. In the following section, we will discuss the results considering previous research and provide conclusions.

## 5. Discussion and Conclusions

By conducting a survey among enterprises in the forest-wood sector and comparing answers based on enterprise size, we were able to evaluate not only differences between micro-enterprises and SMEs but also within micro-enterprises with less than two employees and those with two or more.

We confirmed that SMEs give more importance to reaching new customer groups (H1c) than both categories of micro-enterprises, while for the improving existing products strategy (H1a), we were able to confirm this for those with 0–1 employee but not for those with 2–9 employees, as they value this strategy at least as importantly as SMEs. Regarding customer-specific solutions (H1d), the difference goes in a different direction than we assumed: SMEs give more importance to this strategy than both categories of micro-enterprises, which is unexpected given their predisposition to flexibility [31] and orientation towards customers [18–20]. Based on available data, we could not confirm our assumptions regarding enterprise differences in giving importance to the strategy of introducing entirely new products (H1b) and low-price strategy (H1e).

Except for supporting activities for processes (H2e), we were not able to confirm our hypothesis that SMEs are adopting more innovations than micro-enterprises. For production processes (H2c), the data were not sufficient to confirm it; while for innovations in goods (H2a), services (H2b) and distribution methods (H2d), we found that SMEs are more innovative than micro-enterprises with 0–1 but not more than those with 2–9 employees, which does not correspond to the positive linear association between enterprise size and innovation adoption that was found in most studies [22–24], but supports the finding that the association is not necessarily linear [25].

On the one hand, we were able to confirm that micro-enterprises are more likely to develop innovations by adapting or modifying processes originally developed by other enterprises or organisations (H3c) than SMEs but only for those with 2–9 employees, while those with 0–1 employee have done so less than SMEs. Moreover, the difference is significant only for services, while for goods and processes, we were not able to establish it based on this data. Similarly, we were not able to identify significant differences for innovations introduced by themselves (H3a), in collaboration with other enterprises or organisations (H3b) and completely by other enterprises or organisations (H3d).

Although one of the previous studies showed that micro-enterprises more often adapt innovations from other firms in the sector [3], we did not find any differences in the introduction of products that are new to the market (H4a) and new to the enterprise (H4b). Thus, our findings can neither support the idea that innovations of smaller enterprises can be more radical [31], nor that their innovations are usually only new to them [3,26].

Based on the data from our survey, we were also not able to confirm assumptions about differences between enterprises of different sizes, neither for abandoned or suspended (H5a) and ongoing (H5b) innovation activities nor in any of the listed types of innovation activities (H6a, H6b, H6c, H6d, H6e, H6f, H6g).

In contrast to our expectations, a higher share of SMEs co-operate with suppliers of equipment, materials, components, or software (H7b) than both categories of micro-enterprises. Regarding other types of co-operators (H7a, H7c, H7d, H7e, H7f, H7g, H7h), we could not draw any conclusions based on our survey data, contrary to indications of openness to collaboration in the literature [28–30,32,33]. We also did not find support for the idea that they collaborate less with universities, research institutes and similar institutions [34,35].

Regarding innovations with environmental benefits, we were able to confirm that SMEs have more of these than micro-enterprises only for reduced energy use or $CO_2$ footprints (H8b); while for benefits in recycled waste, water, or materials for their own use of sale (H8f), SMEs surpass only those micro-enterprises with 0–1 employee but not those with 2–9 employees. Possibly, this can be explained with cost issues being a bigger barrier for smaller than larger enterprises [37]. However, no differences could be established for other types of benefits (H8a, H8c, H8d, H8e, H8g, H8h).

In summary, our findings demonstrate that micro-enterprises with 2–9 employees can be more innovative than SMEs. Excluding SMEs from official innovation surveys deprives users of insight into the various innovation activities of this vast segment of enterprises. By including at least larger micro-enterprises in surveys, their representativeness will be improved and allow researchers to conduct better research on innovation management, which can potentially have an impact on policy and decision makers.

*Limitations and Future Research*

Our study also has important limitations. It is focused only on the forest-wood industry sector in one country, and given the low response rates, we can expect that the survey results have a certain amount of bias. There is also the question of the representation of different sectors within the different size categories. Future research should expand this approach to more sectors and countries and try to achieve better response rates that allow comparisons between enterprises of different sizes and sectors.

Moreover, due to changes made to the CIS 2018 questionnaire compared to earlier versions, the findings could not be compared to the CIS data for the same reference period. In late 2020, we were provided access to the CIS 2018 data that the Slovenian statistical office collected in autumn 2019. Unfortunately, the comparability with our data is limited since several questions were changed significantly between the 2016 and 2018 CIS questionnaires.

Another limitation is that the number of employees is not necessarily the best measure of enterprise size in relation to innovativeness—alternative measures, such as perceived size in relation to market potential could be considered [22]. In addition, future research should consider that the relationship between size and innovation is complex and might be attributed to other enterprise characteristics, such as structure [44], organisational culture [44,45], employee's competencies and attitudes [45] and internal communication [46].

**Supplementary Materials:** The following supporting information can be downloaded at: https://www.mdpi.com/article/10.3390/su14041991/s1, Document S1: Survey Questionnaire; Document S2: Survey Invitation; Document S3: Information Sheet with Frequently Asked Questions; Document S4: Comparison of enterprises of different sizes for selected indicators.

**Funding:** The study was funded by the European Commission within the InnoRenew project [Grant Agreement #739574] under the Horizon 2020 Widespread-Teaming program, the Republic of Slovenia (investment funding from the Republic of Slovenia and the European Regional Development Fund) and the Slovenian Research Agency within the project Using questionnaires to measure attitudes and behaviours of building users (Z5-1879).

**Institutional Review Board Statement:** Ethical review and approval were waived for this study as it did not include any ethical issues, and the questionnaire did not collect any personally identifiable data.

**Informed Consent Statement:** Informed consent was obtained from all subjects involved in the study. The survey's introduction emphasized that participation is voluntary, and the respondents could withdraw from participation at any time during the process. The participants were also informed that the data would be used in research publications, for education purposes, and for future research, including by third parties.

**Data Availability Statement:** The data presented in this study is available via the Slovenian Social Science Data Archive [42].

**Acknowledgments:** The author would like to thank enterprises that participated in the study for their collaboration and colleagues at InnoRenew CoE, Mike Burnard for content feedback and Liz Dickinson for language editing.

**Conflicts of Interest:** The author has no competing interest to declare that are relevant to the content of this article.

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
