# Peer review of "Underrated Innovativeness of Micro-Enterprises Compared to Small to Medium Enterprises in the Slovenian Forest-Wood Sector"

_sustainability, doi:10.3390/su14041991_

Round 1

Reviewer 1 Report

The theme of the article is interesting, however the article needs further improvements before being considered for publishing.

The theoretical framework needs to be strengthen, including more international references, and recent ones. There is no elaboration about the innovation concept and theories. The article need to present the innovation of the research, the gap that is filling in this research, the contributions.

The methodology needs to be clarified, and the research design explained in a better way.

The data analysis can be more stronger using inferential statistics, it is important to explicit the validation process of the data collection instrument and also the data.

The conclusions need to be reformulated in order to be linked to the results, and to the theoretical approach. 

Author Response

 Response to Reviewer 1 Comments

Thank you for taking the time to read the article and provide constructive feedback.

Point 1: »The theoretical framework needs to be strengthen, including more international references, and recent ones. There is no elaboration about the innovation concept and theories. The article need to present the innovation of the research, the gap that is filling in this research, the contributions.«

More references were added, and the literature review is now separate from the introduction section, which now includes the research gap. The literature review now includes a theoretical framework and more international and recent references.

Point 2: »The methodology needs to be clarified, and the research design explained in a better way.«

Additional clarifications about the research design were added to different parts of the Methods section.

Point 3: »The data analysis can be more stronger using inferential statistics, it is important to explicit the validation process of the data collection instrument and also the data.«

The data analysis is already based on inferential statistics (Chi-Square test). The overview of results is presented in Attachment 4. The data collection instrument is a replication of questions used in the Community Innovation Survey that is in line with the Oslo Manual guidelines for collecting and interpreting innovation data and is validated by the national statistical office(s). In addition, the instrument was pre-tested – I added that at the end of the second paragraph of the Methods section. Data validity procedures are now described in the last paragraph of the Methods section.

Point 4: »The conclusions need to be reformulated in order to be linked to the results, and to the theoretical approach.«

The conclusions were already linked to results and the literature review – now they are updated based on new references.

Reviewer 2 Report

  1. Title of the article. I think that just "comparison" is a bit downgrading the article. I would suggest to find more pertinent title for the article.
  2. Introduction. Introduction is merged with literature review and the development of hypotheses. I would suggest to develop far further the literature review and to structure it as "review arguments - hypothesis 1, arguments - hypothesis 2, and etc.). I agree that there is limited scope of literature on micro enterprises, but even when rather plausible arguments can be found in literature on small and big companies (tendencies).
  3. Discussion. Discussion is missing (with reference to secondary literature).
  4. Results. Do not end the part of the article by Figure (Figure 8). Some text should be added.
  5. Overall. The article is good, just need some polishing.

Author Response

Response to Reviewer 2 Comments

Thank you for taking the time to read the article and provide constructive feedback.

Point 1: »Title of the article. I think that just "comparison" is a bit downgrading the article. I would suggest to find more pertinent title for the article.«

The title was changed to “Innovation activities of micro and small to medium enterprises in the Slovenian forest-wood sector”.

Point 2: »Introduction. Introduction is merged with literature review and the development of hypotheses. I would suggest to develop far further the literature review and to structure it as "review arguments - hypothesis 1, arguments - hypothesis 2, and etc.). I agree that there is limited scope of literature on micro enterprises, but even when rather plausible arguments can be found in literature on small and big companies (tendencies).«

The literature review is now separate from the introduction and includes several additional references. It is structured according to different question topics, and the hypotheses are now better supported by the literature review.

Point 3: »Discussion. Discussion is missing (with reference to secondary literature).«

The discussion is merged with the conclusions. The text was updated based on new references that were added.

Point 4: »Results. Do not end the part of the article by Figure (Figure 8). Some text should be added.«

The results in Figure 8 are interpreted above it, so it is not necessary to add an additional paragraph after it.

Author Response

Response to Reviewer 3 Comments

Thank you for taking the time to read the article and provide constructive feedback.

Point 1: »“The "introduction" section is too long. There should be an additional section (e.g., literature review) into which the authors will move some of the "introduction" section.

The literature review is now a separate section that includes a big part of the previous introduction section and additional references.

Point 2: »The new section should be supplemented with the characteristics of innovation and the problem of adaptation of innovations based on recent literature. It is necessary to expand the theoretical part and add at least 10 items,

e.g., https://doi.org/10.3390/su12208630 ; https://doi.org/10.1016/S0148-2963(00)00152-1 ;

https://doi.org/10.1007/s10488-013-0486-4 ; https://doi.org/10.3390/su14010140«

The new section includes the definition of innovation and innovation activities and several additional references.

Point 3: »Please adapt "References" to MDPI style. «

The style of references and citations has been changed to ACS according to the MDPI guidelines.

Point 4: »In the last section, please separate the section: "Limitations and future research. «

Limitations and future research are now a separate subsection.

Round 2

Reviewer 1 Report

The authors addressed to my concerns.

Author Response

Thank you.

Reviewer 2 Report

  1. Please do not finish a part of the article with the Figure (the same previously mentioned Figure 8).
  2. Concerning the title. Do not think that changed title is the best possible option. What if "Underrated innovativeness of ..." or similar... 

Author Response

Thank you for the suggestion. I changed the title to "Underrated innovativeness of microenterprises compared to small to medium enterprises in the Slovenian forest-wood sector" and added a paragraph after Figure 8. 

Reviewer 3 Report

The authors have greatly improved the quality of the manuscript

Author Response

Thank you.